# In situ quantification of osmotic pressure within living embryonic tissues

Antoine Vian [1,2], Marie Pochitaloff [1,2], Shuo-Ting Yen[1,2], Sangwoo Kim[1], Jennifer Pollock[1], Yucen Liu[1], Ellen M. Sletten [3] & Otger Campàs [1,2,4,5] ✉

Mechanics is known to play a fundamental role in many cellular and developmental processes. Beyond active forces and material properties, osmotic pressure is believed to control essential cell and tissue characteristics. However, it remains very challenging to perform in situ and in vivo measurements of osmotic pressure. Here we introduce double emulsion droplet sensors that enable local measurements of osmotic pressure intra- and extra-cellularly within 3D multicellular systems, including living tissues. After generating and calibrating the sensors, we measure the osmotic pressure in blastomeres of early zebrafish embryos as well as in the interstitial fluid between the cells of the blastula by monitoring the size of droplets previously inserted in the embryo. Our results show a balance between intracellular and interstitial osmotic pressures, with values of approximately 0.7 MPa, but a large pressure imbalance between the inside and outside of the embryo. The ability to measure osmotic pressure in 3D multicellular systems, including developing embryos and organoids, will help improve our understanding of its role in fundamental biological processes.

Mechanics is known to affect fundamental biological processes across scales, from cellular function to organ formation and tissue homeostasis[1–5]. Actomyosin force generation, cell-cell adhesion, traction forces, and membrane tension have all been shown to affect cellular activity at subcellular and cellular scales[5]. At a multicellular level, active force generation[2,6,7] and spatiotemporal control of tissue material properties[8,9] have been shown to play a key role in tissue morphogenesis during embryonic development, as well as in the control of cell migration[10] and cell differentiation[11,12]. Other fundamental cellular and developmental processes, such as the control of cell and nuclear sizes[13–15], cell division[16], cytoskeletal mechanics[17,18], the emergence of a blastocoel in early mammalian embryos[19], the formation of complex lumen structures during organogenesis (liver, pancreas, lung, etc.)[20–22], and the emergence of gradients in extracellular spaces during embryonic development[9,23], all depend on a tight control of the osmotic pressure both inside cells and in the extracellular space[24,25]. Yet, measuring osmotic pressure remains very challenging,

especially in 3D multicellular systems such as living tissues or organoids, hindering our understanding of the role that osmotic pressure plays in living organisms.

Previous measurements of the hydrostatic pressure difference across the cell surface in animal cells in vitro, or in externally accessible lumens in vivo, have been achieved using either microneedles as a pressure gauge or other surface contact probes, such as atomic force microscopy[16,26–29]. These techniques require an external probe to be in constant contact with the sample, which is invasive and not well-suited for 3D multicellular systems that continuously change shape. Intracellular osmotic pressures in animal, fungal, and plant cells have been estimated in vitro by applying osmotic shocks, with values ranging between 0.1–1 MPa[17,18,30]. Previous microdroplet-based techniques have been developed to measure mechanical stresses[31] or material properties[32] in situ and in vivo, but these do not allow measurements of (osmotic) pressure. Gel microbeads can perform measurements of isotropic stress associated with cellular crowding in multicellular

[1]Department of Mechanical Engineering, University of California, Santa Barbara, CA, USA. [2]Cluster of Excellence Physics of Life, TU Dresden, 01062 Dresden, Germany. [3]Department of Chemistry and Biochemistry, University of California, Los Angeles, CA, USA. [4]Max Planck Institute of Molecular Cell Biology and Genetics, Dresden, Germany. [5]Center for Systems Biology Dresden, 01307 Dresden, Germany. ✉e-mail: otger.campas@tu-dresden.de

systems, but cannot measure osmotic pressure either[33,34]. Finally, measurements of the interstitial fluid osmolality in early zebrafish embryos were achieved using standard osmometers by collecting large interstitial fluid quantities in whole tissue explants[35]. These measurements provided an average value of interstitial fluid osmolality for the entire explant, which required the destruction of the sample, thereby precluding any measurements of spatial or temporal variations in osmotic pressure in the tissue. Therefore, measuring osmotic pressure locally in situ and in vivo, within cells or tissues of developing embryos (including lumen formation in organogenesis), or in other 3D multicellular systems, such as organoids, remains challenging.

Here, we introduce osmotic pressure sensors able to quantify osmotic pressure intra- and extra-cellularly within 3D living tissues, including developing embryos. The sensors consist of double emulsion microdroplets made of a biocompatible oil droplet containing a smaller aqueous droplet with a calibrated concentration of osmolyte (Fig. 1a). The oil surrounding the inner aqueous droplet acts as a protective shell while simultaneously allowing surfactant-mediated water transport, effectively behaving as a water permeable layer (Fig. 1a). By controlling the osmolyte concentration in the inner droplet, as well as the surfactants in the oil and the relative inner/outer droplet sizes, it is possible to generate osmotic pressure sensors with well-defined characteristics. After calibrating the sensors, we used them to measure the osmotic pressure in blastomeres (cells) of early zebrafish embryos, as well as in the interstitial fluid between the cells of the zebrafish blastula. Our results show that double emulsion droplets enable in situ and in vivo measurements of osmotic pressure, both intra- and extra-cellularly within living embryonic tissues.

## Results

### Double emulsion microdroplets as osmotic pressure sensors

Double emulsion droplets, composed of an aqueous droplet embedded in an oil shell (water-in-oil-in-water, or W/O/W double emulsions), could potentially be used as osmometers since water flux through the oil shell is possible in the presence of surfactants[36–38] (Fig. 1a). This water permeability of double emulsion droplets has previously enabled droplet size control by osmotic pressure tailoring[38], the tuning of optical properties in encapsulated colloidal photonics[37], the controlled assembly of colloidal crystals[39], the control of reaction timing in microreactors[40], and also used to generate microgel particles[41]. The water transport through the fluorocarbon oil layer is thought to rely on inverse micelles formed by the surfactant within the oil layer[42–45]. Thanks to the outer water-permeable oil layer, the inner aqueous droplet can increase or decrease its volume as water enters or leaves the droplet, respectively (Fig. 1a). Previous studies have shown that changes in osmolarity in the external medium can drive water flows through the oil shell of the double emulsion droplet[36,38,42,46], indicating

that the system is sensitive to osmotic pressure differences. In order for double emulsion droplets to be used as osmotic pressure sensors, the osmolarity and size of the inner aqueous droplet, as well as the size and surfactant composition of the outer oil layer must be controlled, enabling the generation of stable and calibrated double emulsion droplets.

To produce monodispersed, stable water-in-oil-in-water double emulsion droplets, we used droplet microfluidics[47] (Fig. 2a–c; Methods), as it enables control over the initial volumes of both the inner aqueous droplet and outer oil layer within our desired range (10–40 μm in outer droplet radius). To ensure biocompatibility, we used fluorocarbon oils for the oil phase and non-ionic fluorinated surfactants (Krytox-PEG) to stabilize the droplets (Methods), which have both previously been used in biological applications[48], including as in vivo mechanical stress sensors and actuators[9,31,32,49]. In addition, we used a fluorinated fluorophore[50] to visualize the oil layer using fluorescence microscopy (Methods). Finally, in order to control the osmotic pressure of the inner aqueous droplet, we introduced high molecular weight polyethylenglycol (PEG; a small fraction of it being fluorescently-labeled) as a water soluble non-ionic osmolyte during the generation of droplets (Fig. 2a, b; Methods). Microfluidic generation of such droplets in a polyvinyl alcohol (PVA) solution of fixed osmolarity led to stable double emulsions with controlled initial volumes (Fig. 2c; Methods). The fluorescent dyes in the inner droplet and in the oil layer enable the quantification of inner/outer droplet sizes at high resolution using fluorescence confocal microscopy (Fig. 2d).

Once produced, we characterized the response of double emulsion droplets to controlled changes in osmolarity in the external medium. Placing double emulsion droplets in an aqueous medium containing a salt (NaCl) concentration of 0.4 M drove a progressive and strong reduction in droplet volume as water left the inner droplet through the oil layer (Fig. 2d). This led to an increase in the fluorescent intensity signal in the inner droplet as fluorescent PEG became more concentrated. Monitoring the reduction in the volumes of both inner and outer droplets (Fig. 2a; Methods) showed that both decreased equally over time from their respective initial volumes, eventually reaching equilibrium volumes as the pressure in the inner droplet equilibrated with the external pressure (Fig. 2e). Throughout this process, the oil shell volume remained constant (Fig. 2e), as expected for fluorocarbon oils with low water solubility[42,51], indicating that monitoring the inner or outer droplet volume provides the same information about droplet sizes. Long-term imaging of double emulsion droplets for 12 h shows no changes in the droplet equilibrium volume (Supplementary Fig. 1), indicating that only water is transported through the oil shell. Moreover, imaging droplets for the same period at varying laser intensities displayed a laser power dependent decay in fluorescent PEG signal intensity, indicating that the observed slight decay in fluorescence intensity is mostly due to photobleaching

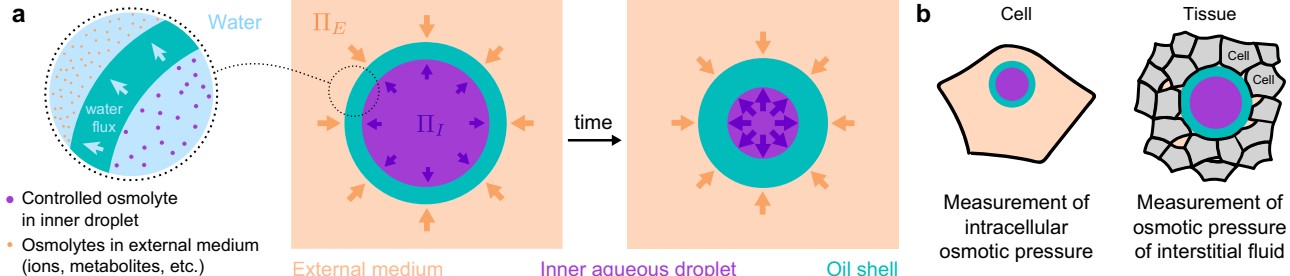

**Fig. 1 | Double emulsion droplets as osmotic pressure sensors. a** Sketch of double emulsion droplets used as osmotic pressure sensors in cells or in the interstitial space between cells within living tissues. Relevant physical parameters are defined. **b** Sketch of a double emulsion droplet inside a cell (left) and in the extracellular space between cells (right), enabling measurements of the intracellular osmotic pressure and of the osmotic pressure of the extracellular interstitial fluid, respectively.

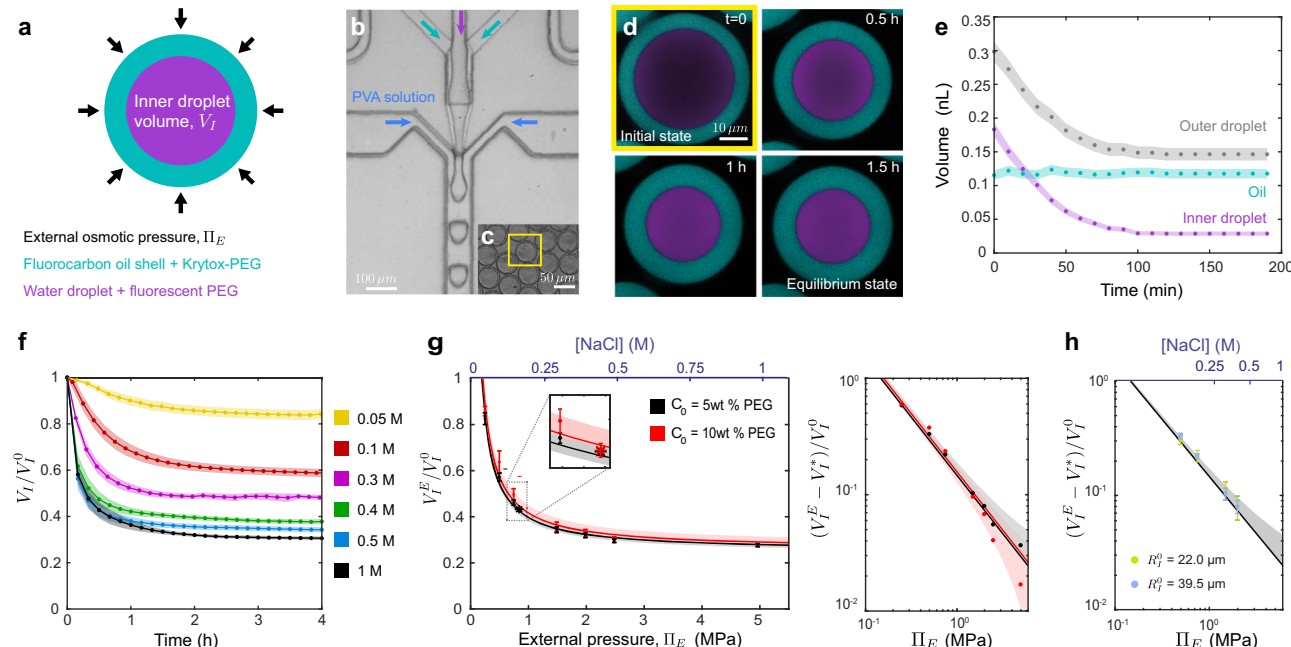

**Fig. 2 | Characterization of double emulsion droplets at equilibrium. a** Sketch of a double emulsion droplet indicating its composition and characteristics. Microfluidic generation (**b**) of double emulsion droplets (**c**). **d** Confocal section of a droplet in a 0.4 M NaCl solution over time showing the temporal reduction in droplet sizes. Fluorocarbon oil (cyan) and fluorescent PEG (purple) are shown (color code as sketched in **a**). **e** Temporal evolution of the inner droplet volume, $V_I$ (purple), the outer droplet volume, $V_T$ (gray) and the oil layer volume (cyan). Error bands are droplet segmentation errors. Representative case, $N = 1$. **f** Temporal evolution of the inner droplet volume, $V_I$ (normalized by the initial volume, $V_I^0$), for double emulsion droplets placed in NaCl solutions of varying osmolarities (Methods). $N = 20$ (yellow), 16 (red), 16 (purple), 15 (green), 21 (blue), 17 (black) droplets for **f**, **g**. Mean ± SD for **f**–**h**. **g** Measured dependence of the equilibrium inner droplet volume, $V_I^E$ (normalized by $V_I^0$), on the externally imposed osmotic pressure, $\Pi_E$,

with initial PEG concentrations, $c_O = 5\%$ w/w (black circles) and 10% w/w (red circles). Linear scale, left panel; log-log scale, right panel. Black and red lines are fits of Eq. 1 to the data with associated confidence bands (68%). Measured equilibrium volumes of the inner droplet for droplets with $c_O = 5\%$ w/w (black asterisk) and 10% w/w (red asterisk) when placed in cell culture media of known osmolarity. $N = 13$ (black), 25 (red) droplets. Small inset is a magnified region of **g**. **h** Initial size dependence of, $(V_I^E - V_I^*)/V_I^0$, on $\Pi_E$ for droplets of initial radius, $R_I^0$ (large droplets, $R_I^0 = 33.5 \pm 0.6$ µm, blue, $N = 26$ (0.5 MPa), 23 (0.75 MPa), 18 (1.5 MPa), 23 (2 MPa); small droplets: $R_I^0 = 12.2 \pm 0.3$ µm, green, $N = 26$ (0.5 MPa), 23 (0.75 MPa), 18 (1.5 MPa), 23 (2 MPa)) but same $c_O$ (5% w/w). Black line is the calibration curve (fit in **g**) for $c_O = 5\%$ w/w. CB (68%) is shown. Source data are provided as a Source Data file.

rather than PEG leakage from the inner droplet (Supplementary Fig. 2), in agreement with the constant equilibrium droplet volume at long timescales. These results indicate that double emulsion droplets have the necessary characteristics to be used as proper osmotic pressure sensors.

To test the sensitivity of double emulsion droplets to different external osmotic pressures, we monitored the temporal evolution of their inner droplet volume $V_I$ when placed in salt (NaCl) solutions of different concentrations, ranging from 0.05 M to 1 M (Fig. 2f). The osmolality of each of these solutions was measured using a commercial osmometer, allowing us to obtain the osmotic pressure $\Pi_E$ of each solution (ranging from 0.25 to 4.96 MPa; Methods; Supplementary Fig. 3). These salt solutions of known osmolalities (and osmotic pressures) were used to calibrate the double emulsion droplets. For all concentrations, the inner droplet volume decreased over time from its initial volume $V_I^0$ until reaching an equilibrium volume $V_I^E$ that depended on the externally imposed osmotic pressure $\Pi_E$, with larger osmotic pressures leading to smaller equilibrium volumes (Fig. 2f). Measurements of the inner and outer droplet interfacial tensions (Supplementary Fig. 4) allow an estimation of the droplet capillary stresses (both approximately of 1 kPa), which are several orders of magnitude smaller than the measured osmotic pressures and, consequently, do not affect our measurements. The equilibrium volume of the inner droplet showed a power law dependence on the external pressure (Fig. 2g), albeit never becoming smaller than a minimal volume, $V_I^*$, associated with PEG volume exclusion (osmotically inactive volume), as previously reported[17,18]. This power law relation is consistent with the inner droplet's osmotic pressure $\Pi_I = A/(V_I - V_I^*)$

being equal to the external osmotic pressure at equilibrium, namely

$$\Pi_E = \frac{A}{V_I^E - V_I^*} \tag{1}$$

where $A$ is a constant associated with the inner droplet osmolyte concentration and can be related to the initial conditions of droplet preparation by $A = \Pi_I^0 \left( V_I^0 - V_I^* \right)$, with $\Pi_I^0$ being the osmotic pressure of the initial PVA solution, fixed in our experiments at 79 mOsm/kg or 0.2 MPa (Methods). Double emulsion droplets with different initial PEG concentrations in the inner droplet also follow Eq. 1 (Fig. 2g). To test if this same relation holds in the presence of more complex external chemical environments, we placed double emulsion droplets in cell culture media (Methods). The resulting equilibrium inner droplet volumes follow the same relation in cell culture media as for simple salt solutions with the same osmotic pressure, regardless of the initial PEG concentrations in the inner droplet (Fig. 2g, small inset). Finally, the same behavior was also observed for different initial inner droplet volumes at fixed PEG concentration in the inner droplet (Fig. 2h). These results indicate that the power law relation in Eq. 1 constitutes a robust calibration curve of double emulsion droplets, providing the relation between the measured inner droplet volume and the osmotic pressure in the external medium at equilibrium.

Beyond equilibrium values, to evaluate the temporal resolution of the measurements, it is important to know the relaxation timescale $\tau_R$ of pressure equilibration in double emulsion droplets. To that end, we monitored the volume of the inner droplet over time and measured

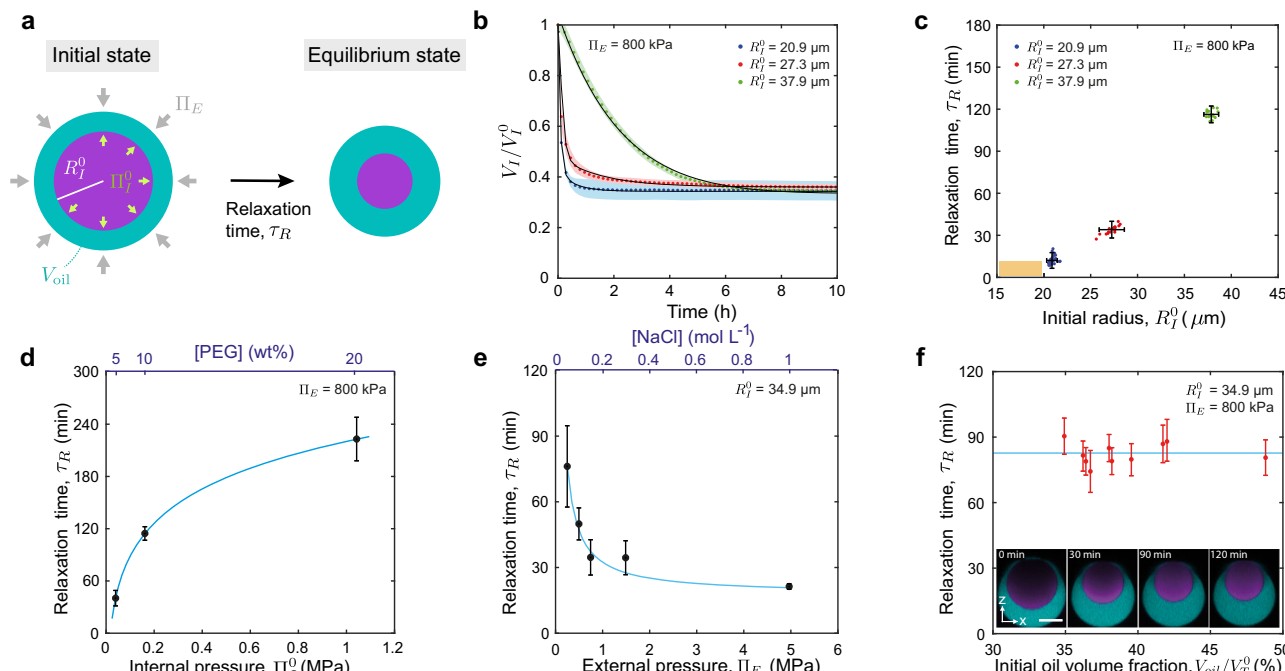

**Fig. 3 | Pressure equilibration timescales of double emulsion droplets. a** Sketch showing a double emulsion droplet of initial inner pressure $\Pi_I^0$ and volume $V_I^0$ (or radius $R_I^0$) and initial oil volume $V_{oil}$, reducing its volume to the equilibrium values over a timescale $\tau_R$. **b** Inner droplet volume relaxation (normalized to the initial volume $V_I^0$) for double emulsion droplets of different initial sizes: $R_I^0 = 37.9 \pm 0.7 \,\mu m$ (green, $N = 20$); $R_I^0 = 27.3 \pm 0.4 \,\mu m$ (red, $N = 20$); $R_I^0 = 20.9 \pm 0.3 \,\mu m$ (blue, $N = 47$). Initial PEG concentration (5% w/w) and fixed $\Pi_E$. Mean ± SD (represented by an error band). Black lines are exponential fits to the data (Methods). **c–f**, Dependence of the measured equilibrium relaxation timescale, $\tau_R$, on the initial inner droplet size, $R_I^0$ (**c** initial PEG concentration (5% w/w) and fixed $\Pi_E$; $N = 20$ (green), 20 (red), 47 (blue) droplets), the initial internal pressure, $\Pi_I^0$ (**d** fixed $R_I^0$ and $\Pi_E$, $N = 13$ (5% w/w), 18 (10% w/w), and 20 (20% w/w) droplets), the externally imposed osmolarity, $\Pi_E$ (**e** initial PEG concentration (5% w/w) and fixed $R_I^0$, $N = 11$ (0.25 MPa), 15 (0.5 MPa), 12 (0.75 MPa), 14 (1.5 MPa) and 16 (5.0 MPa) droplets), and the initial oil volume fraction, $V_{oil}^0/V_T^0$ (**f** initial PEG concentration (10% w/w), fixed $R_I^0$ and $\Pi_E$; $N = 10$ droplets), with $V_T^0$ being the initial total droplet volume. Inset in (**f**) shows z-x imaging plane of a droplet relaxing to the equilibrium state (fluorocarbon oil, cyan; fluorescent PEG, purple). Continuous blue lines in **d** and **e** are fits to the data with the form: $y = ax^b + c$. Scale bar, 25 μm. Mean ± SD for **c–e**. Mean ± error from exponential fit for **f**. Source data are provided as a Source Data file.

the dependence of the relaxation timescale on the different control parameters (Fig. 3a, b; Methods), namely the initial inner droplet radius, $R_I^0$, initial internal pressure, $\Pi_I^0$, imposed external pressure, $\Pi_E$, and the initial oil volume fraction, $V_{oil}/V_T^0$ (with $V_{oil}$ being the volume of the oil shell and $V_T^0$ the initial volume of the entire droplet). The relaxation timescale $\tau_R$ displayed a strong dependence on the initial inner droplet size $R_I^0$, with increasing relaxation time for increasing droplet sizes (Fig. 3c). This behavior is compatible with a power law dependence of the relaxation timescale $\tau_R$ on $R_I^0$ (Supplementary Fig. 5). While smaller values of the initial inner pressure $\Pi_I^0$ led to shorter relaxation timescales (Fig. 3d), pressure equilibration occurred faster for larger external pressures $\Pi_E$ (Fig. 3e). Finally, no dependence of the relaxation timescale on the oil volume fraction $V_{oil}/V_T^0$ was observed (Fig. 3f), likely because there is always a region where the oil layer thickness is small due to the inner droplet buoyancy (Fig. 3f, inset). The measured values of $\tau_R$ were not affected by the presence of fluorinated dye in the fluorocarbon oil (Supplementary Fig. 6). In order to perform measurements of osmotic pressure on relatively short timescales (~10 min), the initial radius of the inner droplet $R_I^0$ should be smaller than approximately 20 μm and have small initial internal pressures (<100 kPa; Fig. 3c). In what follows, we generate droplets with these characteristics ($R_I^0 < 20\,\mu m$ and $\Pi_I^0 < 100$ kPa; Fig. 3c) to perform osmotic pressure measurements in living embryos.

## In situ and in vivo measurements of osmotic pressure in zebrafish blastomeres

After calibrating double emulsion droplets, we performed proof-of-principle experiments to measure the osmotic pressure inside blastomeres (cells) of developing zebrafish embryos (Fig. 4a, b). A single double emulsion droplet was microinjected into the only cell in the

embryo at the 1-cell stage (Methods; Fig. 4c), as previously established[9,31,32]. To measure the local value of the osmotic pressure, we monitored the volume of the inner droplet over time for over 3 h, from the 4-cell stage until the cell size became approximately twice the droplet size (Fig. 4c, d). The measured intracellular osmotic pressure values were of 280 mOsm/kg (0.7 MPa) on average (Fig. 4j) and remained largely constant throughout the measurement period (Fig. 4d). These values were similar to those estimated from osmotic shocks in vitro for the intracellular osmolarity of cells in culture conditions[18] (280–300 mOsm). The measured intracellular osmotic pressure should change to the osmotic pressure of the external medium (E3 buffer; Methods) upon dissolution of cell membranes, since the double emulsion droplet would progressively be exposed to the external embryo E3 medium (Fig. 4j; Methods). We used 2% (w/w) sodium dodecyl sulfate (SDS) to dissolve cells' membranes and completely disperse their contents in the external medium (Fig. 4e; Methods). The osmotic pressure was monitored during the process and found to progressively decrease from its measured intracellular value to the osmotic pressure of the external embryo E3 medium in the presence of 2% w/w SDS (Fig. 4f,j), which was found to be approximately 5-fold smaller than the intracellular osmotic pressure. These results indicate that double emulsion droplets accurately measure the local osmotic pressure, and that cells (blastomeres) in early embryos tightly regulate their intracellular osmotic pressure through division cycles (cleavages).

## Osmotic pressure of interstitial fluid in developing zebrafish embryos

Beyond intracellular osmotic pressure, the osmotic pressure of the extracellular interstitial fluid located between the cells (Fig. 4i) has

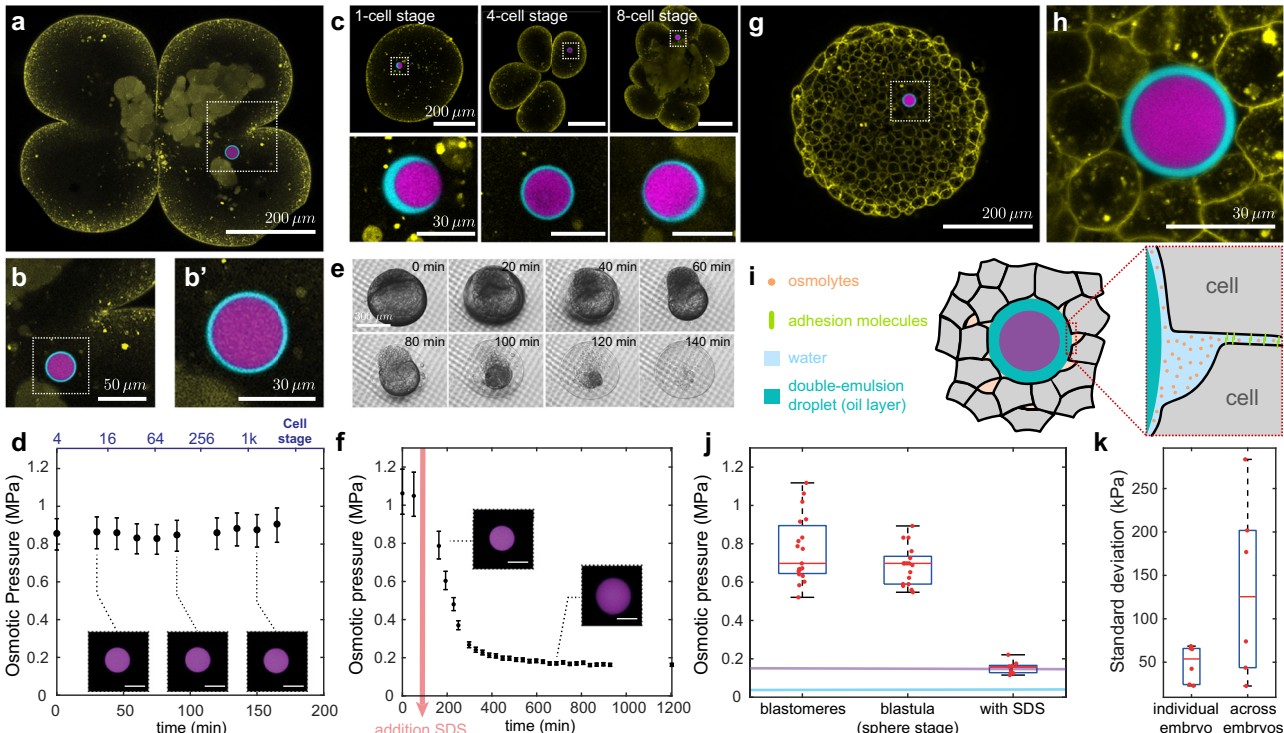

**Fig. 4 | In vivo and in situ measurements of osmotic pressure in blastomeres and in the interstitial fluid of zebrafish embryos. a** Confocal section of a *Tg(actb2: mem-NeonGreen)^hm37* zebrafish embryo transitioning from the 2- to 4-cell stage (membranes, yellow) with a droplet (fluorescent PEG in inner droplet, purple; fluorocarbon oil, cyan) located in one of the blastomeres (cells). **b–b'**, Close ups of the droplet in **a. c** Top panels: Confocal images of a droplet inside a cell of a developing zebrafish embryo at different developmental stages. Bottom panels: close ups of the droplet at each stage. **d** Representative example of measured time evolution of the intracellular osmotic pressure in a developing zebrafish embryo. $N = 1$. Mean ± measurement error bands (obtained by error propagation from calibration curve Fig. 2g) for **d**, **f. e** Timelapse of a zebrafish embryo in 2% w/w SDS solution imaged in an inverted microscope (transmitted light) and sustained on a porous membrane (Methods). **f** Representative time evolution of the osmotic pressure during SDS treatment (2% w/w SDS). $N = 1$. Insets show inner droplet

equatorial confocal sections at different timepoints. **g** Confocal section of a zebrafish embryo blastula at sphere stage (4 hpf; same color code as in **a**) with a droplet inserted in the interstitial fluid between the cells. **h** Close up showing the equatorial confocal section of the droplet in **g. i** Schematic representation of the droplet in between adhering cells and the presence of osmolytes in the interstitial fluid. **j** Measured osmotic pressure inside blastomeres, between the cells (interstitial fluid) of the zebrafish blastula and after SDS treatment. $N = 18$, 20 and 10 droplets (1 droplet per embryo), respectively. Osmotic pressure of E3 buffer (embryo medium) with (violet line) and without (blue line) 2% w/w SDS, measured with a commercial osmometer (Methods). **k** Measured osmotic pressure variation (SD) of temporal readings in individual embryo (SD of temporal readings; $N = 1$) and across the different embryos (SD; $N = 6$). Boxplot show Median, 25th and 75th percentiles, whiskers extend to extreme data points. Source data are provided as a Source Data file.

also been shown to play an important role in morphogenetic events. To measure the osmotic pressure of the interstitial fluid, we injected one double emulsion droplet in between the cells of the zebrafish embryo blastula at sphere stage (Fig. 4g, h; Methods) and monitored the volume of the inner droplet (Methods). We imaged the droplet every 20 min to allow proper equilibration of the droplet volume. Moreover, we checked that the interstitial fluid has enough time to accommodate this change (Supplementary Fig. 7; Methods). The measured osmotic pressure of the interstitial fluid was found to be approximately 700 kPa, corresponding to an osmolality of 280 mOsm/kg on average, nearly identical (within error) to the measured intracellular value of osmotic pressure in blastomeres (Fig. 4j). This value is very close to the average osmolality of 260 mOsm/kg measured in whole zebrafish blastula explants[35], showing that our in vivo and in situ readings are consistent with these previous ex vivo observations. While we could not measure the intracellular osmotic pressure of cells of the blastula at sphere stage because of their small size, our measurements suggest that the intracellular osmotic pressure and the osmotic pressure of the interstitial fluid are constantly balanced, as a mismatch in such high osmotic pressures could be fatal for cells. In contrast, the pressure of the external embryo E3 medium is 17-fold lower than both the interstitial fluid pressure and the intracellular pressure (Fig. 4j).

## Discussion

This work shows that double emulsion droplets can be used as non-invasive, precise and robust osmotic pressure sensors to locally measure osmotic pressure in vivo and in situ within 3D multicellular systems, such as developing embryos, both intra- and extra-cellularly. Using calibrated double emulsion droplets, we quantified the osmotic pressure inside cells as well as in the interstitial fluid between the cells of living zebrafish embryos.

The measured values of osmotic pressure reported herein are in agreement with previous in vitro inferences from osmotic perturbations of animal cells in culture conditions[17,18], and are similar to those values estimated from plasmolysis in plant or fungal cells[30]. A previous measurement of the interstitial fluid osmolarity in zebrafish tissue (blastula) explants[35] reported similar values to those obtained in our measurements. However, those experiments required the destruction of the sample and could only obtain an average value of osmolarity for the entire tissue explant. Our local, in situ measurements show that intracellular osmotic pressure is constant throughout the first divisions in zebrafish embryos, and that osmotic pressures inside cells and in the extracellular spaces (interstitial fluid) are balanced. However, we find that the osmotic pressures in the embryo are 17-fold higher than that of the medium external to the embryo, showing that embryos are able to maintain a large osmotic pressure difference between their

interior and the surrounding environment. It is possible that the vitelline membrane surrounding the embryo mechanically supports this large pressure difference, similarly to cell walls in bacterial or plant cells, which feature similar intracellular osmotic pressures as those reported here for cells of zebrafish embryos. These results suggest that osmolarity is highly regulated both in cells and in the interstitial fluid, in agreement with our observations showing a much smaller variability in osmotic pressure within a given embryo than across different embryos (Fig. 4k).

Since double emulsion osmotic pressure sensors must equilibrate their volume to read the local osmotic pressure, the time resolution of these measurements is limited. For measurements in vivo and in situ with droplets of approximately 30 μm in diameter, the equilibration timescale is less than 10 min. Faster equilibration times are possible for smaller droplets, enabling faster measurements. The timescale of equilibration can also be affected by the ability of fluid to move in the tissue. In the measurements reported above, fluid is able to move between cells at timescales shorter than droplet volume equilibration. However, the equilibration timescale of the droplet could potentially become limited by fluid availability in its neighborhood in tissues with very small interstitial spaces.

Oil droplets have previously been used to measure cell-generated mechanical stresses in vivo and in situ by monitoring the droplet deformations[9,11,31,52]. In this study we employed droplets with high interfacial tensions that do not allow cells in zebrafish embryos to deform them. However, it is a priori possible to use double emulsion droplets to measure simultaneously the local osmotic pressure and the anisotropic mechanical stresses in the tissue by simultaneously monitoring the volume of the inner droplet and the shape deformations of the outer droplet.

Previous in vitro studies have shown that several cell behaviors, from cell migration[26,53] to cortical cell mechanics[17] as well as cell and nuclear size[13], depend on the cell's osmotic pressure. Other works have shown the existence of spatial gradients in extracellular spaces that lead to gradients in tissue stiffness during posterior body axis elongation in zebrafish[9]. Gradients in extracellular spaces can strongly affect tissue hydraulics[23,25,54,55], which depends on the local regulation of interstitial fluid osmolarity. Finally, the formation of the blastocoel and other embryo cavities, as well as the formation of lumen during organogenesis depend directly on the control of osmotic pressure in these structures[19,24,27,54]. The ability to locally measure osmotic pressure in 3D multicellular systems opens new avenues to study its role in all these cellular and developmental processes, during both embryogenesis and in disease states.

## Methods

### Microfluidic device fabrication
The microfluidic devices for producing double emulsions were made of poly(dimethyl siloxane) (PDMS Sylgard 184, Sigma Aldrich, Cat# 761036) and fabricated using soft lithography[56,57]. The template for the double emulsion droplet microfluidic device is based on an existing design[47]. The dimensions of the device were adjusted to achieve the desired droplet sizes. Specifically, we used two different flow focusing devices with different dimensions of the main channel, namely 100 μm width and 60 μm height for the large one and 30 μm width and 30 μm height for the small one. The size of the droplets generated depended on the channel geometry[58,59]. Surface activation of the PDMS devices was done with plasma treatment (Plasma Harrick PDC-32G). Then, a solution containing a cationic polymer, 2% w/w pollydiallyldimethylammonium chloride (PDADMAC, Sigma Aldrich, Cat# 409014) and 1 M NaCl (Sigma, Cat# S9888), was used to render the main channel downstream of the 3D junction hydrophilic. A solution of 2% v/v trichloro(1H,1H,2H,2H-perfluorooctyl) silane (Sigma-Aldrich, Cat#

448931) was used to obtain fluorophilic injection channel upstream of the 3D junction. Double emulsion droplets were produced with nine different microfluidic PDMS devices cast from the same mold.

### Double emulsion droplet composition
The inner droplet was composed of an aqueous solution containing either 5%, 10%, or 20% w/w poly(ethylene glycol) (PEG, Sigma, $M_w$ = 6 kDa, Cat# 81260), corresponding to osmolalities of 16 mOsm/kg, 65.33 mOsm/kg and 420 mOsm/kg (Fig. S3), respectively, and 0.01% w/w of mPEG-Rhodamine (*Creative PEG Works*, Mw = 5 kDa; Cat# PSB-2264) or mPEG-Fluorescein (*Creative PEG Works*, Mw = 5 kDa; Cat# PSB-2254). The concentration of PEG in the inner droplet defines the internal osmotic pressure of the droplet, as PEG cannot go through the oil layer (Supplementary Fig. 1). The presence of PEG also facilitates the generation of double emulsion droplets because it increases the solution viscosity. mPEG-Rhodamine was added at a much smaller concentration to enable droplet size measurements using confocal microscopy. The oil layer surrounding the inner droplet was composed of a fluorinated oil, namely hydrofluoroether (HFE) Novec™ 7700 (3 M; ID 7100094084), containing a fluorinated surfactant Krytox-PEG (RAN BioTech Cat# 008) at a 2% w/w concentration, which is a triblock surfactant that has two perfluorinated blocks that are separated by a PEG-based block[60,61]. For imaging purposes, 0.025 mM of custom-made fluorinated dye $F_{86}Cy5$[50] was added to the oil phase.

### Production of water-in-oil-in-water double emulsion droplets
Each phase was injected in the flow focusing microfluidic device[47], with each flow rate (Fig. 2b; inner flow rate, $Q_i$, magenta; oil flow rate, $Q_m$, cyan; outer flow rate, $Q_o$, blue) being independently controlled by a different syringe pump (New Era Pump System Model #1000). In addition to the phases described above for the inner droplet and the oil layer, the external aqueous phase to generate the droplet contained 10% w/w partially hydrolyzed poly(vinyl alcohol) (PVA, Sigma, $M_w$ = 13-23 kDa, Cat# 363170). Water-in-oil-in-water double emulsion drops with diameters ranging from 25 to 120 μm were formed using the two flow focusing devices described above. Control over the general size of the droplet was achieved by two parameters: the type of device and outer flow rate. For droplets with initial diameter ranging from 60 to 120 μm, we used the large device and outer flow rate $Q_o$ ranging between 3000 to 6000 μL/h. For droplets with initial radius between 20 to 60 μm, we used the small device and outer flow rate $Q_o$ ranging between 300 to 1700 μL/h. To change the initial oil volume fraction, $V_{oil}/V_T^0$, we used a large device and kept $Q_o$ constant at 4500 μL/h, while the ratio $Q_i/Q_m$ was adapted from 1:1 to 8:3.

### Osmotic pressure calibration
The osmolality of all aqueous solutions used for calibration and testing was measured with an osmometer (Advanced instruments, Model 210, Case # 34458). Conversion from osmolality, $\pi_{osm}$, to osmotic pressure, Π, was done using the Van't Hoff Law for dilute solutions, namely Π = $\pi_{osm}RT$, with $R$ the gas constant and $T$ the temperature in Kelvin[17,18].

In order to relate the internal volume of the droplet to the external pressure we produced droplets with initial inner radius of 33.5 ± 0.6 μm. Those droplets were subsequently placed into NaCl solutions with calibrated concentrations of 0.05 M, 0.1 M, 0.15 M, 0.3 M, 0.5 M and 1 M. For ionic and dilute solutions, the osmotic pressure is related to the concentration as follow. The osmolarity $\pi_{osm} = n\phi c$, with $n$ being the number of particles in which the compound dissociates, $\phi$ being the degree of dissociation of the solute and $c$ the solute concentration. In the case if NaCl, $n = 2$ and $\phi = 1$. Knowing $n$, $\phi$, and $c$, for the NaCl solution, we obtained $\pi_{osm}$ and the osmotic

pressure according to $\Pi = \pi_{osm}RT$. This provided a solution of well-known osmotic pressure that was used to calibrate the droplets.

## Storage and osmotic pressure calibration

All droplets produced with microfluidic devices were generated and initially stored in 10% w/w PVA aqueous solution with osmolality of 79 mOsm/kg (or 200 kPa in osmotic pressure).

The osmolality of cell culture media measured in the osmometer was 839 kPa or 338 mOsm/kg. E3 embryo media was composed of NaCl (290 mg/L), KCl (13.33 mg/L), CaCl$_2$ (4.83 mg/L), MgCl$_2$ (81.5 mg/L) and methylene blue (1 vol%, 100 μL/L). The measured osmolality of the E3 embryo media was 11 mOsm/kg (27.3 kPa), which increased to 48 mOsm/kg (118 kPa) when SDS was added at a 2% w/w concentration.

The cell culture media used in calibration experiments was composed of RPMI 1640 (ThermoFisher, Cat# 11875093), supplemented with 1% w/w Penicillin-Streptomycin (ThermoFisher, Cat# 15140122) and 10% w/w Heat Inactivated Fetal Bovine Serum (ThermoFisher, Cat# MT35011CV).

## Characteristic relaxation time

In order to obtain the characteristic timescale of pressure equilibration, we monitored the droplet volume changes over time and fitted an exponential decay to the data. The characteristic relaxation timescale is simply the timescale of the exponential fit.

## Zebrafish husbandry and fish lines

Zebrafish (*Danio rerio*) were raised and bred according to as described previously[62]. Animals were raised and experiments were performed following all ethical regulations and the protocols approved by the Institutional Animal Care and Use Committee (IACUC) at the University of California, Santa Barbara (protocol number 886). A *Tg(actb2:mem-NeonGreen)*$^{hm37}$ transgenic line was used for ubiquitous labeling of cell membrane of zebrafish embryos. Sex experiments were not necessary, as zebrafish embryos at the studies stage have not yet undergone sex determination.

## Injection of double emulsion droplets in zebrafish embryos

Zebrafish embryos at 1-cell stage were chemically dechorionated by 1 mg/mL of pronase (Roche, Cat# 10165921001) in E3 buffer. Embryos at sphere stage were dechorionated manually. Embryos were all microinjected with double emulsion droplet in 0.1 M KCl (Sigma, Cat# P3911) solution using a picolitre injector (Warner Instruments LLC, PLI-100A). Micropipettes for microinjection were made from microcapillaries (World Precision Instrument; TW100F-4) using a Sutter P-1000 needle puller and were coated with 2% w/w PDADMAC in 1 M NaCl to avoid rupture of the double emulsion droplet inside the micropipette. The diameter of the inner droplets of the double emulsions ranged between 20–35 μm. Double emulsion droplets were backloaded into the microneedle, which tends to accumulate at the tip of the needle due to gravity. Injection pressure was tuned to achieve the injection of single droplets in the embryo.

## Mounting and imaging of double emulsion droplets in zebrafish embryos

All images were acquired using a Zeiss LSM710 laser scanning confocal microscope. Imaging of zebrafish embryos injected with double emulsion droplets were mounted in 0.75% low-melting point agarose (Invitrogen; Cat# 16520-050) mixed with 25% OptiPrep™ density gradient medium (Sigma; Cat# D1556) in E3 buffer (without methylene blue)[63] in a glass-bottom dish (MatTek; P35G-1.5-14-C) with two layers of silicone isolators (Electron Microscope Sciences, Cat.# 70336-61). For SDS treatment, a 40-μm nylon mesh, which was cut out from cell strainers (Fisher Scientific; Cat# 22363547), was used to provide a porous seal instead of a cover slide. SDS treatment was administered at 128-cell stage of the zebrafish development and measurement of the drop size was manually performed every 15 min for 4 h.

Images of early development zebrafish were taken using a 10x air objective (EC Epiplan-Neofluar 10x, NA 0.25, Carl Zeiss Inc.). For measurements of volume changes in double emulsion droplets, 3D time-lapses of droplets were acquired using a 40x water immersion objective (LD C-Apochromat 40x, NA 1.1 W, Carl Zeiss Inc.) at 25 °C. Confocal section in z were between 5–10 μm with the 10x objective and 1–2 μm for the 40x objective.

## Imaging of double emulsion droplets for calibration purposes

3D confocal timelapses of double emulsion droplets were acquired on a Zeiss LSM710 laser scanning confocal microscope with a 40x water immersion objective (LD C-Apochromat 1.1 W, Carl Zeiss Inc.) at room temperature.

## Analysis of inner and outer droplet volumes

To quantitatively obtain the droplet's size from imaging data, we developed a custom-made Matlab code[64]. First, maximum intensity projections (MIP) of the measured z-stack of multiple droplets in a region of interest were obtained for the inner droplet at every timepoint. We focused on the inner droplet because changes in outer droplet volume follow changes in inner droplet volume (Fig. 2e). Individual timelapses of inner droplets' MIPs were segmented by thresholding a grayscale image with an input threshold value. Segmentation artifacts smaller than a critical object size are removed and binary erosion operation and binary dilation operation are applied consecutively to generate smooth droplet interfaces. Individual droplets were then labeled at each time point and tracked over time based on the shortest distance criterion between consecutive time points. For each droplet identified in the segmented image, the droplet area $A$ was computed by counting number of pixels. Since the inner droplets maintained spherical shape, the inner droplet volume $V_I$ was obtained from

$$V_I(t) = \frac{4}{3}\pi\left(\sqrt{\frac{A(t)}{\pi}}\right)^3, \tag{2}$$

## Fluorescence recovery after photobleaching (FRAP)

A solution of Dextran-Alexa Fluor 488 (10,000 MW) in DI water at a concentration of 10 mg/mL was prepared, and approximately 0.5 nL of this solution was injected in the interstitial fluid of a zebrafish embryo at the sphere stage following the same protocol as for injection of droplets (see above). After 20 min, the zebrafish embryos were mounted for imaging as described above. Imaging was done as described above and using a 25x water objective. To measure the fluorescence recovery after photobleaching, a Region of Interest (ROI) of 30 μm by 30 μm was defined and the fluorescence signal within this region was photobleached by illuminating it with a 488 nm laser (80% laser power; 10 frames). Fluorescence intensity was then monitored in the defined ROI for 15 min after photobleaching. The average intensity in the ROI was measured and fitted with a single exponential function to obtain the recovery timescale. The Fiji plugins from Stowers ImageJ Plugins were used for the analysis of the recovery timescale[65].

## Statistics and reproducibility

We did not use statistical methods to predetermine sample size in droplets or experiments involving zebrafish embryos, however, our sample sizes are similar to those reported in previous publications[9,32,66]. No samples were excluded from the analysis. Analysis of all the images was done by automated software to ensure blinding and avoid biases in the analysis. No randomization

of the data was used. The characteristic reduction of volume and subsequent stabilization of the double emulsion droplets submitted to osmotic pressure was independently observed 86 times (Fig. 2d). Observations reported in Fig. 2b, c were reproducible and observed 20 times. Observations of droplets in zebrafish embryos at different stages (Fig. 4a, g) were reproducible and observed 50 times.

## Reporting summary

Further information on research design is available in the Nature Portfolio Reporting Summary linked to this article.

## Data availability

All data supporting the findings of this study are available within the article and its supplementary files. Any additional requests for information can be directed to, and will be fulfilled by, the corresponding author. Source data are provided with this paper.

## Code availability

The custom-made image analysis code used in this article is publicly available on GitHub[64] at: https://github.com/campaslab/Osmotic-Pressure-droplet-volume-reading.

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

## Acknowledgements

We thank I. Lim (UCLA) for sharing custom fluorinated dyes, S. Megason (Harvard Medical School) for providing *Tg(actb2:mem-NeonGreen)^{hm37}* embryos, A. Kickuth (Brugués Lab, MPI CBG, Dresden) for sharing her mounting method of early zebrafish embryos, K. Mantoani for SDS screening of zebrafish embryos, and G. Stooke-Vaughan and A. Boutillon for their help with FRAP experiments. We also thank all members of the Campàs lab (especially C. Gomez) for discussions and technical support, the CNSI microfluidics and cleanroom facilities at UCSB as well as the UCSB Animal Research Center for support. A.V. was supported by a postdoctoral fellowship from the Swiss National Foundation (P400PB_191065). This work was supported by the National Institute of General Medical Sciences of the National Institutes of Health (R01GM135380 to E.S. and O.C.), and the Deutsche Forschungsgemeinschaft (DFG, German Research Foundation) under Germany's Excellence Strategy—EXC 2068—390729961—Cluster of Excellence Physics of Life of TU Dresden.

## Author contributions

A.V. and O.C. designed research; A.V., M.P., S.-T.Y. performed experiments; A.V. and S.K. analyzed the data; S.K. developed image analysis codes; J.P. and Y.L. provided technical assistance with microfluidics; E.S. provided technical expertise and fluorophilic dyes; A.V. and O.C. wrote the paper; O.C. supervised the project.

## Funding

## Competing interests

A.V. and O.C. declare the following competing interests: Provisional patent, application number 63/383,647, Systems and Methods for Measuring Osmotic Pressure, Antoine Vian and Otger Campas, 2022. There are no other competing interests for any of the authors.
