## [Peer Review File · Nature Communications]

REVIEWER COMMENTS

Reviewer #1 (Remarks to the Author):

This manuscript reports the microfluidic production of water-in-oil-in-water double-emulsion droplets as a sensor to quantify osmotic pressure intra- and extra-cellularly within living embryonic tissues. The middle perfluorocarbon oil layer contains a biodegradable fluorosurfactant, which acts as a selective membrane that only allows water to diffuse. By calibrating the concentration of osmolyte in the core, droplet size, core-to-shell size ratio, as well as the surfactants in the oil phase, the osmotic pressure sensor is calibrated to achieve equilibrium size and relaxation time scale for measuring osmolarity of early zebrafish embryos. The measured osmotic pressure by double-emulsion sensor is in agreement with the previously reported values done by invasive methods. Overall, this is a well-written and well-organized manuscript. Although the principle employed for the osmotic pressure measurement, inflation or deflation of double-emulsion droplets, has been well known, I believe this work makes a valuable contribution to understanding the role of osmotic pressure in biological processes in situ and in vivo, especially for multicellular systems. The following comments will help the authors improve the work before publication.

1. Is there any possibility for Laplace pressure to influence the droplet size at equilibrium? It would be great if the authors discuss the relative magnitude of the Laplace pressure to osmotic pressure and the influence.
2. How long do the double-emulsion droplets stay stable during in vivo and in situ measurement?
3. During blastula stage, the double-emulsion drops were inserted in the interstitial fluid of cells. Was there a critical point at which the double-emulsion drop ruptures because it could not attain to the pressure from the growing cells?
4. In the interstitial fluid, the amount of continuous phase is very limited. At the same time, the size of the double-emulsion droplets is comparable to the cell size, does the measured osmolarity on the interstitial fluid is measured at sufficient time for the double-emulsion drops to reach the equilibrium?
5. Please check whether the symbol on Figure 2f represents 0.1 M, or whether it corresponds to the value written on experimental section (0.05 M, 0.1 M, 0.15 M, 0.3 M, 0.5 M, and 1 M)
6. Please attach legend to the supplementary figures.
7. It is suggested to discuss how osmotic pressure has been used in double-emulsion droplets. For example, see Langmuir 2012, 28, 9944–9952, Langmuir 2013, 29, 38, 11849–11857, Small 2014, 10, 1155–1162, and many others.

Reviewer #2 (Remarks to the Author):

In the manuscript entitled 'In situ quantification of osmotic pressure within living embryonic tissues', Vian et al. described a novel approach to measure osmotic pressure in living tissues, at both the intracellular and extracellular levels. This is a highly original and much needed approach for the field of developmental biology and mechanobiology in general, particularly in view of the emerging importance of fluid pressure in regulating diverse developmental processes. Overall, the methods are well described, main texts written well and the cited references are fairly comprehensive. I do have a few questions for the authors to clarify, before consideration for publication:

Major:

1. Fig. 4d - why does the authors not report data for 1 or 2 cell stage?
2. Fig. 4d - in the case of 4-cell stage, the blastomere osmotic pressure is 0.85 MPa, while the external medium osmotic pressure is around 0.2 MPa (Fig. 4j after SDS treatment). This would imply an osmotic pressure difference across the 4-cell stage blastomere of 0.65 MPa, which implies a hydrostatic pressure of blastomere cytoplasm to be ~650 kPa at equilibrium (when hydrostatic pressure equals osmotic pressure difference across the cortex). Typical values of cell pressure is often in the range of hundreds of Pascals - how do the authors explain this? Or did I miss something?
3. Related to the above questions, can the authors perform pipette aspiration to measure surface tension of the blastomere, so as to directly infer the cytoplasmic hydrostatic pressure (Laplace law), which can in turn validate the measured osmotic pressure in the blastomeres using beads?
4. Fig. 4j - it isn't clear from the main texts and figure legend what stage of blastomeres the authors refer to for the quantified intracellular pressure. Also, why is the average pressure (0.7MPa) different from that of Fig. 4d (>0.8 MPa)?

Minor:

1. In the introduction, in describing the use of micro needles to study lumen pressure in vivo, it will be relevant to cite recent work Chan et al. 2019 DOI: 10.1038/s41586-019-1309-x.

2. Fig 2f - There seems to be a typo in the figure texts, I presume the different colours indicate different osmolality, but they all indicate 0.1M?

Reviewer #3 (Remarks to the Author):

The article by Antoine Vian and colleagues focuses on developing a tool to measure osmotic pressure in cells and tissues. They took advantage of microfluidic made double emulsion droplets that contain a PEG solution surrounded by a layer of fluorocarbon oil, the emulsion being stabilized by fluorinated surfactants. The paper explains the principle, the techniques, and the characterization of the osmotic probes. The authors successfully show:

1-the double-emulsion is responding efficiently to osmotic shocks

2-the dynamics of volume change and relaxation to equilibrium are dependent on initial volume of the drop, and intensity of the osmotic shock.

3-the osmotic values reported by the probe in cells and tissues are fully compatible with gross measurements reported elsewhere, establishing a proof of concept in biological living samples.

The paper is well written, concise and clearly explain. It brings a new tool that as a lot of potential in many open questions in tissue mechanics such as cavity formation, cell sorting and of course shape and volume variations. It is a clear candidate for Nature Communications, and I only have a few remarks to help the authors strenghtening their point.

1-In cells or tissues, droplets may be subjected to compressive and extensile forces. This group has used similar mechanical probes to successfully measure stresses in tissues. I was wondering if the authors have observed deformation of the droplets? Could the droplets report on both osmotic pressure and stress fields at the same time?

2-A similar question could be about the hydrostatic pressure. If burried inside a cavity in the tissue, contractile forces within the tissue may generate hydrostatic pressure onto the probe that could change the volume of the aqueous phase, impacting the osmotic pressure measurement. My intuition is that hydrostatic pressure coming from tissue forces is several orders of magnitude lower than osmotic pressure. But this should be clarified and justified somewhere.

3-In complex solutions, like cell media, it is difficult to be absolutely sure that none of the solutes actually cross the oil layer, as some amphipathic molecules could still diffuse inside the droplets. The impact on osmotic pressure is probably very small, but the effect will be stronger on droplets with low initial osmolarity. A way to test this is probably to form droplets with 0 Osm aqueous phase, or very small osmolarity, and see if on the long time, any divergence to the equilibrium value is seen. This would indicate any long term divergence that should be corrected in the long time experiments (tissues).

format comments:

1-the colour code should be indicated on the figures in fig 2g, 2h, 3b and 3c.

2-in fig 2g, it would be better to have a log-log plot in both graphs to show the power law dependence. In this case, fig 2h should be changed accordingly.

3-in fig 3c it might be interesting to have relaxation times for 35 and 45 microns, to know if the relation between relaxation time and initial radius is linear, quadratic or something else (cubic?). Cubic could be interesting as it would scale with volume...

Reply to Reviewer #1:

This manuscript reports the microfluidic production of water-in-oil-in-water double-emulsion droplets as a sensor to quantify osmotic pressure intra- and extra-cellularly within living embryonic tissues. The middle perfluorocarbon oil layer contains a biodegradable fluorosurfactant, which acts as a selective membrane that only allows water to diffuse. By calibrating the concentration of osmolyte in the core, droplet size, core-to-shell size ratio, as well as the surfactants in the oil phase, the osmotic pressure sensor is calibrated to achieve equilibrium size and relaxation time scale for measuring osmolarity of early zebrafish embryos. The measured osmotic pressure by double-emulsion sensor is in agreement with the previously reported values done by invasive methods. Overall, this is a well-written and well-organized manuscript. Although the principle employed for the osmotic pressure measurement, inflation or deflation of double-emulsion droplets, has been well known, I believe this work makes a valuable contribution to understanding the role of osmotic pressure in biological processes in situ and in vivo, especially for multicellular systems. The following comments will help the authors improve the work before publication.

1. Is there any possibility for Laplace pressure to influence the droplet size at equilibrium? It would be great if the authors discuss the relative magnitude of the Laplace pressure to osmotic pressure and the influence.

We thank the referee for bringing up this point. The scale of the Laplace pressure (or capillary stress) is given by $\sim\gamma/R$, with γ being the interfacial tension and the R the droplet radius. We have now measured the interfacial tensions of the droplet interfaces in the same conditions in which double-emulsion droplets are prepared. We find values of 5.9 ± 0.2 mN/m and 6.1 ± 0.3 mN/m for the interfacial tensions of the outer interface (fluorocarbon oil with fluorosurfactant in di water) and the inner interface (fluorocarbon oil with fluorosurfactant in a PEG solution), respectively (Supplementary Figure 4). Knowing the radius of the droplet (ranging between $R \approx 10 - 50 \mu\text{m}$), the Laplace pressure ranges between 100-600 Pa (or 0.0001-0.0006 MPa). These values are several orders of magnitude smaller than the osmotic pressure (0.1-1 MPa). Consequently, the Laplace pressure plays a negligible role in the determination of the droplet size in equilibrium. We have included the measured interfacial tensions as Supplementary Information (Supplementary Figure 4) and also an explanation of this point in the new version of the manuscript.

2. How long do the double-emulsion droplets stay stable during in vivo and in situ measurement?

In general, we can follow the droplets for as long as we can image it in the tissue. While we could follow the droplet in almost all cases, some droplets moved so deep in the tissue that it was not possible to image them anymore in our confocal microscope. This happened after 20 hours in the case of droplets injected inside blastomeres (cells). For droplets injected in the interstitial tissue, we followed them for up to 5 hours. We have not observed the droplets becoming unstable over these time periods in any of our experiments.

3. During blastula stage, the double-emulsion drops were inserted in the interstitial fluid of cells. Was there a critical point at which the double-emulsion drop ruptures because it could not attain to the pressure from the growing cells?

This is a very interesting point. We have never observed the double-emulsion droplet becoming unstable or rupturing. This suggests that the tissue pressure associated with cell proliferation and cell crowding is much smaller than the osmotic pressure. Indeed, direct measurements of tissue pressure in other systems (mouse mandible, Campas et al., Nature Methods, 2014; cellular aggregates, Mohagheghian et al., Nature Communications, 2018) indicate that the tissue pressure

is less than 1 kPa (or 0.001 MPa), much smaller than the measured values of osmotic pressure (0.1-1 MPa).

4. In the interstitial fluid, the amount of continuous phase is very limited. At the same time, the size of the double-emulsion droplets is comparable to the cell size, does the measured osmolarity on the interstitial fluid is measured at sufficient time for the double-emulsion drops to reach the equilibrium?

The droplets used for osmotic pressure measurements in interstitial fluid equilibrate over timescales of approximately 10 min outside the embryo (Fig. 3c). We inject these droplets 1 hour before starting imaging to let both the droplet equilibrate and the tissue recover from the injection. Once the tissue recovers and the droplet is fully embedded in the tissue, the interstitial spaces in the tissue are small but visible. To directly assess how fast water can move in the interstitial spaces, we injected low amounts of fluorescent Dextran in the interstitial space to visualize them (Supplementary Fig. 7). We then performed Fluorescence Recovery After Photobleaching (FRAP) experiments of the interstitial Dextran and measured the timescale at which Dextran recovers. All our measurements indicate that Dextran recovers in approximately 1 minute. Since Dextran is considerably bigger than water molecules, we expect water to be able to move even faster through the interstitial spaces, so the measured timescale for Dextran is an upper bound for water. Since all our osmotic pressure measurements in the blastula are obtained at 15 minutes intervals, it should not be a problem for water to move through the interstitial fluid at these timescales. In addition, cells allow water to flow in and out of them, so the water content available is not only that in the interstitial fluid, potentially enabling even a faster equilibration. That said, in other tissues with much smaller interstitial spaces and/or if cells prevented water exchange, the double emulsion droplet could take longer to equilibrate.

In the new version of the manuscript, we have added the fluorescent Dextran measurements in the Supplementary Fig. 7 and also a discussion about the timescale of relaxation in different situations.

5. Please check whether the symbol on Figure 2f represents 0.1 M, or whether it corresponds to the value written on experimental section (0.05 M, 0.1 M, 0.15 M, 0.3 M, 0.5 M, and 1 M)

We thank the reviewer for pointing out this typo. We have corrected it in the new version of the manuscript.

6. Please attach legend to the supplementary figures.

We have now included a legend to all supplementary figures.

7. It is suggested to discuss how osmotic pressure has been used in double-emulsion droplets. For example, see Langmuir 2012, 28, 9944–9952, Langmuir 2013, 29, 38, 11849–11857, Small 2014, 10, 1155–1162, and many others.

We have now explained these (and other) previous works on this topic in the new version of the manuscript.

Reply to Reviewer #2:

In the manuscript entitled ‘In situ quantification of osmotic pressure within living embryonic tissues’, Vian etc. described a novel approach to measure osmotic pressure in living tissues, at both the

intracellular and extracellular levels. This is a highly original and much needed approach for the field of developmental biology and mechanobiology in general, particularly in view of the emerging importance of fluid pressure in regulating diverse developmental processes. Overall, the methods are well described, main texts written well and the cited references are fairly comprehensive. I do have a few questions for the authors to clarify, before consideration for publication:

Major:

1. Fig. 4d - why does the authors not report data for 1 or 2 cell stage?

We unfortunately cannot report the pressure at 1- and 2-cell stages because of the time it takes to inject the droplet in the embryo and mount it for imaging. The double-emulsion droplet is injected at the 1-cell stage and the embryo is mounted for imaging immediately after. If maintained at the appropriate temperature (28°C), the embryos are already at the 4-cell stage at the earliest point that we can start imaging. To obtain the image of the droplet in a 1-cell stage embryo (Fig. 4c), we placed the embryo on ice immediately after injection of the droplet to slow down development substantially. While this procedure allows us to slow down the embryo enough to take an image of the droplet at 1-cell stage, we prefer not to report quantitative measurements of the osmotic pressure in these conditions, as we do not know how this temperature change may affect cells and the normal development of the embryo. We therefore reported the osmotic pressure from the 4-cell stage, as this is the earliest stage we can image for embryos in standard conditions (28°C).

2. Fig. 4d - in the case of 4-cell stage, the blastomere osmotic pressure is 0.85 MPa, while the external medium osmotic pressure is around 0.2 MPa (Fig. 4j after SDS treatment). This would imply an osmotic pressure difference across the 4-cell stage blastomere of 0.65 MPa, which implies a hydrostatic pressure of blastomere cytoplasm to be ~650 kPa at equilibrium (when hydrostatic pressure equals osmotic pressure difference across the cortex). Typical values of cell pressure is often in the range of hundreds of Pascals - how do the authors explain this? Or did I miss something?

In Fig. 4j we report the osmotic pressure inside blastomeres and in the interstitial fluid (blastula) measured with double-emulsion droplets. We also show the values of the osmotic pressure inside the embryo measured with the double-emulsion droplet after addition of SDS ('with SDS' in Fig. 4j). In addition, we show the values of the osmotic pressure of the external medium without (light blue horizontal line) and with SDS (violet horizontal line) as measured with an osmometer. The measured value of the osmotic pressure inside the embryo in the presence of SDS ('with SDS' in Fig. 4j) is the same as the osmotic pressure measured in the external medium with SDS (violet horizontal line). In other words, SDS ruptures the cell membranes, as expected, and the pressure inside the embryo reaches the same value as in the external medium.

We agree with the reviewer that there is a transient state (Fig. 4f) just after the addition of SDS in which the osmotic pressures are not equilibrated. SDS progressively breaks down cell membranes and the transient imbalance in osmotic pressure also likely contributes to popping cells (the cell debris after cell popping can be seen in Fig. 4e). Eventually, when cells are ruptured, the osmotic pressure in the embryo becomes the same as in the external medium, as explained in the previous paragraph.

Interestingly, there is a major osmotic pressure difference between the interior of the embryo and the external medium outside the embryo. While the osmotic pressure inside a blastomere and in the interstitial fluid are largely equilibrated (Fig. 4j), the osmotic pressures inside the embryo and outside of it are very different (by at least one order of magnitude). This suggests that some physical structure is mechanically sustaining this pressure difference. We believe this structure is the vitelline membrane, which may play a similar role as the cell wall in plant cells (but here for the entire embryo). In the SDS experiments (Fig. 4e), one can see the spherical vitelline membrane remaining unperturbed as the cells' membranes are broken down by SDS. The embryo has

initially the spherical shape of the vitelline membrane (Fig. 4e, time 0 min), suggesting that it is indeed the vitelline membrane that is sustaining the high osmotic pressure in the embryo.

3. Related to the above questions, can the authors perform pipette aspiration to measure surface tension of the blastomere, so as to directly infer the cytoplasmic hydrostatic pressure (Laplace law), which can in turn validate the measured osmotic pressure in the blastomeres using beads?

We unfortunately cannot use micropipette aspiration of zebrafish blastomeres because, as mentioned above, the embryo is surrounded by a vitelline membrane (made of a dense polymeric fabric) that prevents direct contact of the micropipette and the blastomere (cell) surface. The best cross-validation we have found are measurements of the interstitial fluid osmolarity in the zebrafish blastula at the same stage of our study (exactly same tissue and same stage), which were achieved using standard osmometers by collecting large interstitial fluid quantities in whole tissue explants of many different embryos. The obtained value of interstitial fluid osmolarity in those experiments was 275 mOsm/L, very close to the values 280 mOsm/L of our *in situ* and *in vivo* measurements.

4. Fig. 4j - it isn't clear from the main texts and figure legend what stage of blastomeres the authors refer to for the quantified intracellular pressure. Also, why is the average pressure (0.7MPa) different from that of Fig. 4d (>0.8 MPa)?

In Fig. 4d, we show the time evolution of the osmotic pressure inside one blastomere (with the stages indicated on the upper y axis) and is about 0.8 MPa. Since the osmotic pressure in a single blastomere does not change significantly over time, we defined the average osmotic pressure inside a blastomere (time average). In Fig. 4j, each red dot corresponds to the average pressure of a blastomere in different embryos. The measured ensemble average of those pressures (about 0.7 MPa) corresponds to the average value of osmotic pressure in blastomeres of different embryos. So, while the 0.8 MPa value in Fig. 4d corresponds to the osmotic pressure inside a blastomere of one embryo, the average value of 0.7 MPa in Fig. 4j corresponds to the ensemble average value across embryos.

Minor:

1. In the introduction, in describing the use of micro needles to study lumen pressure *in vivo*, it will be relevant to cite recent work Chan et al. 2019 DOI: 10.1038/s41586-019-1309-x.

We thank the reviewer for pointing this out. We have now added it to the new version of the manuscript.

2. Fig 2f - There seems to be a typo in the figure texts, I presume the different colours indicate different osmolality, but they all indicate 0.1M?

It is indeed a typo. Thank you for pointing it out. We have now corrected it in the new version of the manuscript.

Reply to Reviewer #3:

The article by Antoine Vian and colleagues focuses on developing a tool to measure osmotic pressure in cells and tissues. They took advantage of microfluidic made double emulsion droplets that contain a PEG solution surrounded by a layer of fluorocarbon oil, the emulsion being stabilized by fluorinated surfactants. The paper explains the principle, the techniques, and the characterization of the osmotic probes. The authors successfully show:

- 1-the double-emulsion is responding efficiently to osmotic shocks
- 2-the dynamics of volume change and relaxation to equilibrium are dependent on initial volume of the drop, and intensity of the osmotic shock.
- 3-the osmotic values reported by the probe in cells and tissues are fully compatible with gross measurements reported elsewhere, establishing a proof of concept in biological living samples.

The paper is well written, concise and clearly explain. It brings a new tool that as a lot of potential in many open questions in tissue mechanics such as cavity formation, cell sorting and of course shape and volume variations. It is a clear candidate for Nature Communications, and I only have a few remarks to help the authors strenghtening their point.

1-In cells or tissues, droplets may be subjected to compressive and extensile forces. This group has used similar mechanical probes to successfully measure stresses in tissues. I was wondering if the authors have observed deformation of the droplets? Could the droplets report on both osmotic pressure and stress fields at the same time?

We thank the reviewer for raising this interesting point. Yes, it should a priori be possible to measure simultaneously the osmotic pressure and cell-generated stresses with double emulsion droplets. As the reviewer mentioned, our group developed and established the use of droplets (Campas et al., Nature Methods, 2014) to measure cell and tissue generated forces. However, we have always used that technique in mouse tissues (Campas et al., Nature Methods, 2014; Lucio et al., Scientific Reports, 2017; Parada et al., Dev Cell, 2022). We have tried the same droplets to measure forces in zebrafish but observed no deformations because the forces are smaller in zebrafish tissues than in mouse tissues. In other words, the interfacial tension of these droplets is too high for cells in zebrafish embryos to deform them. Since the double emulsion droplets reported here have the same interfacial tension than these previously used droplets (see Supplementary Fig. 4), cells in zebrafish cannot deform them. This is why we did not observe droplet deformations in double emulsion droplets. Actually, to measure mechanical stresses in zebrafish, we always used magnetic droplets (Mongera et al., Nature, 2018; Kim et al., Nature Physics, 2021; Mongera et al., Nature Materials, 2023) because they have a much lower interfacial tension, allowing cells in zebrafish embryos to deform the droplets.

That said, as the reviewer rightly points out, it should in principle be possible to use the double emulsion droplets developed here to measure both osmotic pressure and cell-generated forces simultaneously, provided that the droplet interfacial tension is low enough to allow cells to deform the droplet. We have added a comment on this in the Discussion section of the new version of the manuscript.

2-A similar question could be about the hydrostatic pressure. If burried inside a cavity in the tissue, contractile forces within the tissue may generate hydrostatic pressure onto the probe that could change the volume of the aqueous phase, impacting the osmotic pressure measurement. My intuition is that hydrostatic pressure coming from tissue forces is several orders of magnitude lower than osmotic pressure. But this should be clarified and justified somewhere.

We agree with the reviewer that this could, in principle, be a possibility. However, as the reviewer rightly suggest, this contribution is much smaller. Direct measurements of tissue pressure in other systems (mouse mandible, Campas et al., Nature Methods, 2014; cellular aggregates, Mohagheghian, Nature Communications, 2018) indicate that the tissue pressure is less that 1 kPa, much smaller than the measured values of osmotic pressure. Moreover, we previously measured the typical values of active cellular forces in zebrafish tissues with magnetic droplets (Mongera et al., Nature, 2018) and are about 100 Pa, several orders of magnitude smaller than the osmotic pressure. In addition, direct measurements of both isotropic and anisotropic cellular stresses in the zebrafish blastula (Mohagheghian, Nature Communications, 2018), exactly the system studied

here, show stresses below approximately 500 Pa. So, contractile cellular forces, and even tissue pressure (the isotropic component of the mechanical stresses in the tissue), have a negligible contribution to the changes in the volume of double emulsion droplets compared to the effect of the osmotic pressure, which is several orders of magnitude higher. We have now explained this point in details in the Discussion section of the new version of the manuscript.

3-In complex solutions, like cell media, it is difficult to be absolutely sure that none of the solutes actually cross the oil layer, as some amphipathic molecules could still diffuse inside the droplets. The impact on osmotic pressure is probably very small, but the effect will be stronger on droplets with low initial osmolarity. A way to test this is probably to form droplets with 0 Osm aqueous phase, or very small osmolarity, and see if on the long time, any divergence to the equilibrium value is seen. This would indicate any long term divergence that should be corrected in the long time experiments (tissues).

We agree with the reviewer that this is in principle possible. We have performed long time experiments (12 hours) using cell culture media to test if the measured equilibrium osmotic pressure is slowly drifting, which would indicate that some molecules (other than water) are getting into the droplet (see Supplementary Fig. 1). For the values of osmotic pressures measured, we do not observe any long timescales drift in the equilibrium value of the osmotic pressure, indicating that if some molecules are diffusing inside the droplets their contribution is negligible.

We have added the new data as Supplementary Fig. 1 and included a discussion on this point in the new version of the manuscript.

format comments:

1-the colour code should be indicated on the figures in fig 2g, 2h, 3b and 3c.

We have now indicated the color code in the mentioned figures.

2-in fig 2g, it would be better to have a log-log plot in both graphs to show the power law dependence. In this case, fig 2h should be changed accordingly.

We completely agree with the reviewer that the power law dependence is important and that log-log plots are the way to show that. We have now added the log-log version of panel 2h for completeness, as suggested by this reviewer.

3-in fig 3c it might be interesting to have relaxation times for 35 and 45 microns, to know if the relation between relaxation time and initial radius is linear, quadratic or something else (cubic?). Cubic could be interesting as it would scale with volume...

From the data we have, it seems that the relation between the relaxation time and initial droplet radius is close to cubic. To obtain this, we plotted our data in log-log scale and fitted a line. The slope of the line corresponds to the exponent of the power law. Our results indicate that the exponent is 3.8 ± 0.5 (Supplementary Fig. 5). However, to properly determine the exact power law exponent we would need sizes of droplets spanning at least 2 decades, which would involve a lot of work in terms of manufacturing new microfluidic devices and gathering this data. Since this is beyond the scope of this manuscript, we think this analysis provides a good initial assessment of the power law exponent. We have included this result in Supplementary Fig. 5 and described it in the main text.

REVIEWERS' COMMENTS

Reviewer #1 (Remarks to the Author):

The authors have made significant improvements to the manuscript, addressing the concerns raised during the initial review. The revisions have significantly improved the clarity and overall quality of the paper. In particular, the author has provided a clearer explanation regarding the negligible role of Laplace pressure in determining droplet size in equilibrium. Furthermore, the potential limitations of the osmotic pressure sensors, arising from in vivo and in situ equilibration timescales of droplets in smaller interstitial spaces, have been thoroughly clarified. These revisions collectively strengthen the manuscript, rendering it suitable for publication in its current form.

Reviewer #2 (Remarks to the Author):

The authors have sufficiently addressed all my previous questions. I therefore recommend publication for the manuscripts in Nature Communications.

Reviewer #3 (Remarks to the Author):

The authors have successfully answered all my concerns, and performed the requested analysis. I support publication without further revision.